# Photo-Induced Depletion of Binding Sites in DNA-PAINT Microscopy

**DOI:** 10.3390/molecules23123165

**Published:** 2018-11-30

**Authors:** Philipp Blumhardt, Johannes Stein, Jonas Mücksch, Florian Stehr, Julian Bauer, Ralf Jungmann, Petra Schwille

**Affiliations:** 1Cellular and Molecular Biophysics, Max Planck Institute of Biochemistry, Am Klopferspitz 18, 82152 Martinsried, Germany; blumhardt@biochem.mpg.de (P.B.); jstein@biochem.mpg.de (J.S.); muecksch@biochem.mpg.de (J.M.); stehr@biochem.mpg.de (F.S.); jubauer@biochem.mpg.de (J.B.); 2Molecular Imaging and Bionanotechnology, Max Planck Institute of Biochemistry, Am Klopferspitz 18, 82152 Martinsried, Germany; jungmann@biochem.mpg.de; 3Department of Physics and Center for Nanoscience, Ludwig Maximilian University, Geschwister-Scholl-Platz 1, 80539 Munich, Germany

**Keywords:** DNA-PAINT, surface-integrated fluorescence correlation spectroscopy (SI-FCS), reactive oxygen species, photo-induced DNA damage, super-resolution microscopy

## Abstract

The limited photon budget of fluorescent dyes is the main limitation for localization precision in localization-based super-resolution microscopy. Points accumulation for imaging in nanoscale topography (PAINT)-based techniques use the reversible binding of fluorophores and can sample a single binding site multiple times, thus elegantly circumventing the photon budget limitation. With DNA-based PAINT (DNA-PAINT), resolutions down to a few nanometers have been reached on DNA-origami nanostructures. However, for long acquisition times, we find a photo-induced depletion of binding sites in DNA-PAINT microscopy that ultimately limits the quality of the rendered images. Here we systematically investigate the loss of binding sites in DNA-PAINT imaging and support the observations with measurements of DNA hybridization kinetics via surface-integrated fluorescence correlation spectroscopy (SI-FCS). We do not only show that the depletion of binding sites is clearly photo-induced, but also provide evidence that it is mainly caused by dye-induced generation of reactive oxygen species (ROS). We evaluate two possible strategies to reduce the depletion of binding sites: By addition of oxygen scavenging reagents, and by the positioning of the fluorescent dye at a larger distance from the binding site.

## 1. Introduction

Super-resolution microscopy has greatly contributed to the study of biological specimens with resolutions down to few nanometers while retaining the high specificity of fluorescent labels [1,2,3]. The stochastic blinking of individual fluorophores enables the precise localization of molecules in various single molecule localization microscopy (SMLM) methods. In many variants of SMLM, in particular the prominent PALM [4,5] and (d)STORM [6,7], the number of photons available from permanently bound, individual fluorophores determines the achievable localization precision [8,9]. In particular, the spatial information from non-functional or immediately photo-bleached labels is entirely lost. In contrast, points accumulation for imaging in nanoscale topography (PAINT) [10] generates the blinking of fluorophores by reversible binding reactions. While an individual binding event is still limited by the photon-budget of the fluorescent dye, binding sites can be revisited by fresh probes and thus contribute to higher resolved images [11]. In DNA-based PAINT (DNA-PAINT) [11,12] the structure of interest is labeled with a short DNA single strand (docking strand), serving as binding site for fluorescently labeled complementary single (imager) strands. The formed duplex immobilizes the imager for the time of binding and creates a bright, localized spot on the detector, usually a sensitive camera, while freely diffusing imager strands remain blurred as a constant background intensity.

Transient binding reactions not only enable super-resolution microscopy, but their kinetics also reflect on the nature of the binding process. Under appropriate imaging conditions, the kinetics of the transient binding can be directly extracted from time traces of the localization data [12,13]. In samples with high densities of binding events, where localization fails, fluorescence correlation spectroscopy (FCS) can reliably extract binding kinetics [14,15,16]. We recently showed that surface-integrated fluorescence correlation spectroscopy (SI-FCS) can determine the kinetic rates of transient DNA hybridization [15].

However, the advantage of DNA-PAINT in that individual binding sites are revisited is limited for long acquisitions, due to photo-induced damages, effectively creating an upper limit for the image quality. In addition, SI-FCS experiments suffer from the depletion of bindings sites, complicating the correct extraction of kinetic rates. Unwanted photo-induced effects are intrinsic to fluorescence microscopy [17,18]. Excited states of fluorescent molecules in general, but in particular long-lived triplet states, are prone to oxidation or reduction and the subsequent generation of highly reactive molecules [18,19,20,21,22]. Triplet states have been found to play a role in photo-bleaching pathways and to promote the generation of reactive oxygen species (ROS) [19,22,23,24,25,26,27,28]. ROS are not only known to bleach fluorescent dyes [18,29] and have phototoxic effects on biological samples [30,31], but also to damage DNA [32,33].

The damage induced by ROS is significantly reduced by the use of oxygen scavenging buffers that remove molecular oxygen from solution and thereby lower the amount of reactive oxygen species. The popular enzymatic oxygen scavenging system glucose oxidase, catalase and glucose (GO+C) [34,35] produces gluconic acid, consequently acidifying the sample. GO+C is thus not suited for long acquisitions or pH dependent systems [36,37,38]. An alternative systems is the combination of protocatechuate-dioxygenase and 3,4-protocatechuic acid (PCD+PCA) [39,40]. The oxidation of PCA produces muconic acid, but at the same time has a buffering effect around pH 8, leading to improved pH stability compared to GO+C [41]. More recently, pyranose oxidase, catalase and glucose (PO+C) have been reported as effective oxygen scavengers with no acidifying effect over the time span of hours [41]. In particular, when studying the DNA duplex formation above the melting temperature, prerequisite for DNA-PAINT imaging, the reaction kinetics are sensitive to the pH of the solution and salt concentrations [12].

Since molecular oxygen acts as triplet quencher, its removal increases the triplet lifetime, leading to long-lived dark states and decreased fluorescence [18,22,42]. The vitamin E analog Trolox has been found to serve as effective triplet quencher in combination with oxygen scavenging systems, resulting in comparably bright and photostable fluorescence imaging conditions [43,44]. In this study, we systematically investigate the stability for long acquisition time series in DNA-PAINT microscopy and SI-FCS kinetic measurements. We provide two practical solutions to the inherent problem of phototoxicity, by comparing two oxygen scavenging reagents, and by presenting a modified imager that utilizes a higher distance of fluorescent dye to docking site.

## 2. Results

To systematically investigate the effect of the photo-induced depletion of binding sites, we compared DNA-PAINT super-resolution microscopy and SI-FCS kinetic measurements from samples in the presence and absence of oxygen scavenging systems. If the depletion of binding sites is caused by ROS, oxygen scavenging buffers will lead to a longer lifetime of binding sites. Additionally, we probed whether the distance of fluorescent dye to the docking strand influences the rate of depletion. To achieve optimal resolution in SMLM, the fluorescent dye is conventionally placed in closest possible proximity to the labeling site. In DNA-PAINT, this is easily achieved by design. In many cases, including this study, the DNA origami scaffold and the fluorescent probe on the imager strand are upon binding only separated by a short spacer [11,12,45]. This arrangement, however, also creates ROS close to the docking strand, and is therefore prone to damaging of DNA bases resulting in an increased depletion rate of binding sites. Assuming an isotropic diffusion of ROS, an increase in the distance between fluorescent dye and the docking strand ought to decrease the probability for interaction. We investigated the depletion of docking sites in the following five conditions (Figure 1): Conventional: we used standard conditions as commonly found in DNA-PAINT super-resolution microscopy and SI-FCS measurements [11,15];Oxygen scavengers:(a)PO+C: we added PO+C as oxygen scavenging system and Trolox as triplet quencher to samples otherwise identical to (1);(b)PCD+PCA: as in (2a), but with PCD+PCA as oxygen scavenging system and Trolox added;18-mer spacer: (a)we extended the 10-nucleotide (nt) imager by an additional 18-mer double stranded spacer sequence to attach the dye at greater distance from the hybridizing docking strand, maintaining otherwise conditions as in (1);(b)we added PO+C as in (2a) to samples otherwise identical to (3a).

Depletion of binding sites in DNA-PAINT manifests itself in a decrease in the number of localizations with increasing measurement time (Figure A1, left panel, raw data is available as Appendix A). To study the underlying effect, two processes have to be disentangled: First, the bleaching of fluorescence imager strands in solution, and second, a depletion of docking sites. The effective concentration of fluorescent imager strands within the total internal reflection fluorescence (TIRF) excitation volume can be assumed equilibrated with the bulk solution concentration, as the exchange rate of imager strands from solution by diffusion is about four orders of magnitude faster than the exposure time or minimum correlation time (Figure A2). Bleaching of the bulk imager solution is usually negligible in DNA-PAINT, as the reservoir of fluorophores in solution is large enough so that the concentration of fluorescent dye is not affected by imaging a small volume via TIRF illumination. Accordingly, moving the sample laterally by more than the size of the illumination fully recovers the number of initial localizations (Figure A1, right panel). Thus, photo-bleaching does not significantly affect the bulk dye concentration. The second effect, the depletion of binding sites, however, affects the surface-immobilized sample, therefore dominates in the observation region and accumulates over time.

To observe the photo-induced depletion of docking sites under conventionally used imaging conditions, we rendered super-resolved DNA-PAINT images from five subsets of one 25,000 frame or 83-min-long acquisition at a peak irradiance of $I_{0}=0.2\;\mathrm{kW}/{\mathrm{cm}}^{2}$, allowing for the localization of individual binding sites. Depending on the particular application, DNA-PAINT experiments are conventionally performed with peak irradiances up to $6\;\mathrm{kW}/{\mathrm{cm}}^{2}$ [11], further enhancing the problem of photo-induced damages. The first subset, covering the first 17 min, renders a super-resolved image with the majority of docking sites visible. Due to the limited incorporation efficiency, not all docking sites are observable in the first subset [46]. Later subsets show a decreasing number of localized binding sites, indicating that less docking strands are available for hybridization of imager strands on the DNA origami scaffold (Figure 2a and Figure A3). Time traces of localizations within circular areas enclosing the individual docking sites show frequent binding events in the beginning of the acquisition, but eventually turn dark during the measurement (Figure 2b).

We further quantitatively investigated the depletion of binding sites in DNA-PAINT microscopy by automated alignment and averaging of the acquired DNA origami nanostructures (Figure A4), as published previously [11,46]. The drift correction, identification and averaging of nanostructures were performed on the complete time series prior to division into the subsets. Selected individual binding sites on the averaged image were then back-translated to individual nanostructures (Picasso: ‘Unfold’) to analyze the intensity trace of each docking site individually. To account for rare unspecific binding events, only traces with more than three localizations per subset were counted as active docking sites. This cutoff slightly changes the absolute numbers of active docking sites, but does not influence the qualitative shape of the observed decays. Following this analysis, roughly 10 docking sites are on average detected as active in the initial subset for the conventional imager in absence (1) and presence of oxygen scavenging buffers (2a,b) (Figure 3a, left panel), in agreement with previous investigations [46]. With increasing measurement time, increasing numbers of docking sites become inactive, leaving on average six binding sites per nanostructure active, with the conventional imager at the end of the time series. In stark contrast, less than one binding site is depleted on average during the entire measurement in presence of oxygen scavenging buffers. The depletion of binding sites is limited to the irradiated sample region, and non-irradiated areas are indistinguishable from a fresh sample (Figure 3a, right panel). In case of the 18-mer spacer (3a,b), we observe a decreased association rate, leading not only to a lower number of localizations, but also to docking sites not being visited frequently enough to classify docking sites reliably as active within one subset. Using the whole time series to identify docking sites, the total number of active docking sites is recovered to about 10, similar to the conventional imager (Figure A5). Irrespective of the low association rate, docking sites are also depleted for the imager carrying the 18-mer spacer ($I_{0}=0.2\;\mathrm{kW}/{\mathrm{cm}}^{2}$) (Figure 3a). For the combination of 18-mer spacer and oxygen scavenger PO+C (3b), we observe an increased error rate within the automated structure alignment and a slightly decreased number of active docking sites (Figure A5). As mentioned, the cut-off for unspecific binding events does not influence the qualitative shape of the decay.

Further, the depletion can also be analyzed based on the kinetics of the reoccurring binding events to individual DNA origami. We use the well-established analysis of the duration of individual binding events and the time span between two consecutive binding events, referred to as bright time $\tau_{B}$ and dark time $\tau_{D}$, respectively [11,12,13]. For the conventional imager strand (1), we observe an increasing dark time for later subsets, which directly reflects on the depletion of binding sites (Figure 3b, top). For completely depleted binding sites, the dark time becomes theoretically infinite. In practice, noise misinterpreted as localization event limits the dark times and leads to a large scatter of the distribution of dark times for later subsets. With the addition of oxygen scavenging buffers (2a,b), the average dark time appears shorter and does not show any significant dependence on the measurement time, indicating a strong reduction in the depletion of binding sites. The two oxygen scavenging systems PO+C and PCD+PCA seem to perform similarly well in reduction of the depletion. For the 18-mer spacer, dark times are higher as for the conventional imager, reproducing the apparent lower association rate (Figure 2b and Figure 3b). Without oxygen scavengers (3a), an increase in dark times is observed, as depletion of binding sites is induced under the chosen irradiation conditions. Adding oxygen-scavenger to the 18-mer spacer (3b) stabilizes the dark times on the level observed in the first subset without oxygen scavengers (3a).

The bright time $\tau_{B}$, in return, is stable or slightly decreases with time (Figure 3b, bottom). The conventional imager and the 18-mer spacer show similar bright times, shortened by the photo-bleaching of the fluorescent dye to less than 2 seconds. With the addition of oxygen scavenging buffers, the bright times increase to roughly 4 seconds, matching the values expected from SI-FCS measurements for a bleaching-free regime (Table 1 from SI-FCS results below). The observed slight decrease in bright time can be attributed to an artifact arising for two simultaneous binding events that are misinterpreted by the localization algorithm [11] as one longer binding event (Figure A6). The more binding sites are accessible, the higher is the probability for two imager strands to bind simultaneously. With increasing measurement time and according depletion of binding sites, the probability to bind two imager strands decreases, and thus the bright time approaches its true value in the later subsets. The dependencies of dark and bright time on the number of docking sites per nanostructure appear similarly in simulations (Figure A6b). Experimentally, the decreasing bright time can be avoided by lower imager concentrations at the cost of extended measurement times to maintain the total amount of localizations. The increasing dark time, however, is intrinsic to the depletion of docking sites and can particularly hinder quantitative analyses based on the association rate [13].

The high irradiances necessary for precise localization of fluorophores in super-resolution microscopy lead to a high probability of photo-bleaching fluorophores while still being attached to the docking strand [15], thus adding an additional layer of complexity. To further separate the effect of photo-bleaching from the depletion of individual binding sites, we analyzed the hybridization kinetics in a low irradiance regime ($I_{0}=0.018\;\mathrm{kW}/{\mathrm{cm}}^{2}$) with surface-integrated fluorescence correlation spectroscopy (SI-FCS) [15]. The fluctuations in fluorescence intensity of transiently hybridizing imager and docking strands are analyzed to extract the binding rates. This approach does not rely on any localization or discrimination of individual binding events and can thus be performed at orders of magnitude lower irradiances. Nonetheless, SI-FCS traces exhibit a decaying mean fluorescence intensity that is accounted for by a monoexponential detrending of the intensity trace before the autocorrelation [15]. The fluorescence intensity I(t) of each integrated region is fitted and subsequently divided by: (1)$$I\left(t\right)=I_{0}\exp\left(-\kappa \;t\right)+I_{\infty },$$
with the effective depletion rate κ characterizing the loss in fluorescence intensity, the amplitude $I_{0}$ and the offset $I_{\infty }$ (Figure A7). For a one-component reversible binding, the SI-FCS autocorrelation function $G_{N}\left(\tau \right)$, with the amplitude normalized to one, is an exponential function decaying with a characteristic decay time $\tau_{c}$:(2)$$G_{N}\left(\tau \right)=\exp\left(-\frac{\tau }{\tau_{c}}\right),$$

A detailed derivation of the SI-FCS autocorrelation function is found in [15]. The monoexponential correction (Equation (1)) allows the accurate quantification of kinetics via SI-FCS, when the characteristic decay time $\tau_{c}$ of the autocorrelation function (ACF) is much shorter than the characteristic time of detrending ($\tau_{c}\ll 1/\kappa$).

With higher average concentration of fluorescent imager in solution 〈A〉, the probability of docking sites to be occupied increases. In SI-FCS, an increase of imager concentration manifests itself as a decrease of the characteristic decay time $\tau_{c}$, which can be expressed in terms of the association and dissociation rates $k_{a}$ and $k_{d}$, respectively:(3)$$\tau_{c}\left(\langle A\rangle \right)=\frac{1}{k_{a}\langle A\rangle +k_{d}}$$

We performed five titration series of the imager concentration 〈A〉 with otherwise identical conditions as described above: with conventional imager in absence of oxygen scavenging buffers (1), with PO+C added (2a), PCD+PCA added (2b), with the 18-mer spacer (3a), and the combination of 18-mer spacer and PO+C (3b). We obtained similar hybridization rates for conventional imager with and without oxygen scavenger system (Figure 4a). Comparing the conventional to the oxygen-scavenged buffer, the dissociation rate is slightly lowered ($\left(0.303\pm 0.010\right)\;s^{-1}$ compared to $\left(0.268\pm 0.017\right)\;s^{-1}$, and $\left(0.25\pm 0.02\right)\;s^{-1}$ for conditions (1), (2a), and (2b) respectively). This decrease is potentially caused by residual photo-bleaching of bound fluorophores, which was previously not observed when determining a bleaching-free regime by variation of the excitation power [15]. The imager with 18-mer spacer (3a,b) shows a decreased association rate (Figure 4b), in agreement with the lower binding frequency observed in DNA-PAINT traces (Figure 2b) and the higher dark times (Figure 3b). A lower association rate can be caused by different steric features of the imager, which are reflected by a larger hydrodynamic radius. We determined the translational diffusion coefficient as $\left(120\pm \;20\right)\;µm^{2}/s$ for the 18-mer spacer, being by a factor 1.7 smaller than the diffusion coefficient of the conventional imager (Figure A2c). We observe a difference for the association rate in absence (3a) and presence (3b) of PO+C only slightly larger than the estimated error of the measurement. The dissociation rate is similar to the conditions in presence of oxygen scavenging buffer (Table 1).

To investigate the effect of depletion of binding sites in SI-FCS, we compared the rate of the detrending κ (Equation (1) and Figure A7a) in absence and presence of oxygen scavenging buffers and the 18-mer spacer. If the process of depletion of docking sites is mediated by the fluorescent dye, only hybridized duplexes create damages. Therefore, κ represents an effective depletion rate, which is given by a depletion rate constant δ specific to the investigated system, lowered by multiplication with the occupation probability ρ(〈A〉) of docking sites:(4)κ(ρ)=δ·ρ(〈A〉),

The occupation probability ρ is intuitively accessible as the fraction of the bright time $\tau_{B}$ to the duration of one binding cycle, represented by the sum of bright and dark time $\left(\tau_{B}+\tau_{D}\right)$:(5)$$\rho =\;\frac{\tau_{B}}{\tau_{B}+\tau_{D}}$$

Bright and dark times are principally accessible from low irradiance PAINT measurements, allowing for the individual detection of docking sites. High irradiances, however, as required to resolve multiple narrow spaced binding sites on one DNA origami nanostructure, reduce the bright time and hinder the direct determination of the occupation probability. Without the need for localization, ρ can be expressed in variables accessible from SI-FCS titrations, in particular the association rate $k_{a}=1/\left(\tau_{D}\langle A\rangle \right)$ and the dissociation rate $k_{d}=1/\tau_{B}$, as:(6)$$\rho =\;\frac{1}{1+k_{d}{\left(k_{a}\langle A\rangle \right)}^{-1}}$$

Experimentally, we determine the specific depletion rate constant δ by varying the concentration 〈A〉 of fluorescent imager in solution (Equation (4)). For the conventional imager (1), we confirm the linear dependence of κ(ρ) with a slope of $\delta_{\mathrm{conv}}=\left(470\pm 40\right)\cdot 10^{-6}\;s^{-1}$. Strikingly, the slope δ decreases by two orders of magnitude when oxygen scavenger is added (Figure 4c), thus nearly eliminating the depletion of binding sites ($\left(20\pm 19\right)\cdot 10^{-6}\;s^{-1}$ and $\left(30\pm 50\right)\cdot 10^{-6}\;s^{-1}$, for PO+C and PCD+PCA, respectively). Similarly, the 18-mer spacer reduces the depletion rate to $\left(7\pm 4\right)\cdot 10^{-6}\;s^{-1}$ and $\left(33\pm 18\right)\cdot 10^{-6}\;s^{-1}$ in absence and presence of PO+C, respectively (Figure 4d). For the low irradiances ($I_{0}=0.018\;\mathrm{kW}/{\mathrm{cm}}^{2}$), employed in SI-FCS, the effect of depletion of binding sites approaches zero for both oxygen scavenging systems and the 18-mer spacer. As apparent from the estimated errors of the individual measurements, the differences observed for the close to zero depletion rates are indistinguishable within the precision of the measurement. The depletion is only distinguishable from zero at high imager concentrations (Figure 4b and Table 2).

## 3. Discussion

To summarize, we have confirmed that docking strands in DNA-PAINT microscopy are depleted by irradiation with visible light in an indirect process mediated by the excitation of fluorescent dyes. DNA-PAINT microscopy with long acquisition series showed that individual docking sites get irreversibly lost in irradiated areas (Figure 2 and Figure 3). In the majority of SMLM methods, the resolution and signal-to-noise ratio (SNR) of super-resolved images are limited by the photon-budget of fluorophores irreversibly bound to labeled sites [8,9]. Making use of the reoccurring binding and a large reservoir of fluorescent imager, DNA-PAINT is less limited by the photo-bleaching of fluorophores to precisely localize binding sites [11]. In this work, we showed that DNA-PAINT is ultimately limited by the eventual damage of docking sites.

We showed evidence that the depletion of docking strands of DNA origami nanostructures is caused by damages of DNA handles and can thus be distinguished from photo-bleaching of fluorophores (Figure 3). In our SI-FCS experiments, we find a linear dependence of the effective depletion rate on the presence of fluorescently labeled imager (Figure 4c,d). This does not only show that the depletion process is mediated by bound fluorophores, but it also indicates that direct photo-damage of DNA is negligible compared to fluorophore-mediated damages. Our experiments in oxygen scavenging buffers strongly indicate the involvement of ROS in the depletion of available docking strands. Not only do we see a significantly reduced depletion in high irradiance DNA-PAINT microscopy ($I_{0}=0.2\;{\mathrm{kW}/\mathrm{cm}}^{2}$) (Figure 2 and Figure 3), but also in low irradiance SI-FCS measurements ($I_{0}=0.018\;{\mathrm{kW}/\mathrm{cm}}^{2}$), we observed a close to zero depletion rate (Figure 4c,d and Table 2). The importance of ROS for photo-induced damages following fluorescence excitation has been studied not only in the context of fluorescence photo-bleaching [18,19,22,27,28,29], but also photo-toxicity [30,31] and in particular DNA damage [32,33,47]. ROS have also been shown to contribute to photo-induced unbinding of proteins [48,49]. Generally, ROS are one of the main sources for DNA damage and have been intensively investigated in relation to several kinds of cancer [50,51,52,53]. Our results show the cause for the depletion to be ROS generated downstream of the fluorescence excitation, diffusing to the DNA origami nanostructures and finally damaging the exposed docking stands. From our experiments, it is not possible to identify the type of created ROS, the type of induced DNA damage or the affected DNA bases. Different ROS, most prominently singlet oxygen and superoxide radicals, are supposedly formed from excited fluorescent states [29,54]. The likelihood and type of induced damage is reported to be sequence dependent and enriched at duplex ends [33]. Previous studies suggest that guanine is preferentially damaged, due to its lowest redox potential among the DNA bases [55,56,57]. Not only were oxidation products frequently found in guanine repeats [33,55], but also electron hole diffusion along the DNA leads to guanine oxidation distant from the site of single electron transfer [58,59,60]. On the other hand, adenine and thymine are the predominant bases on the docking strand that are closest to the fluorescent dye upon imager binding. Thus, most likely, a variety of damage types is induced simultaneously.

The probability of a ROS to reach a docking site by 3D diffusion scales with the squared distance between both, assuming a much longer free path length of the ROS [61,62]. Accordingly, a larger spacing between docking strands and fluorophores reduced the depletion of binding sites (Figure 4b) at low irradiances ($0.018\;\mathrm{kW}/{\mathrm{cm}}^{2}$). Oxygen scavenging buffers and the 18-mer spacer show similar depletion rate constants ($\left(20\pm 19\right)\cdot 10^{-6}\;s^{-1}$, $\left(30\pm 50\right)\cdot 10^{-6}\;s^{-1}$ and $\left(7\pm 4\right)\cdot 10^{-6}\;s^{-1}$, for PO+C, PCD+PCA and 18-mer spacer, respectively), suggesting that both are similarly effective in eliminating photo-toxic effects (Figure 4c,d and Table 2). Addition of PO+C to the 18-mer spacer did not show any further improvement for SI-FCS measurements. The given error of the close to zero depletion rates potentially underestimates the measurement uncertainty, as slow exponential decays (up to $10^{6}\;s$) are fitted only with limited accuracy based on 5-h measurements. Based on our results, we regard them indistinguishable. At the high irradiances (0.2 $\mathrm{kW}/{\mathrm{cm}}^{2}$) used for super-resolution microscopy, the 18-mer spacer is significantly less efficient in lowering the depletion (Figure 3). In contrast, the depletion seems to occur at a rate similar to the conventional (1) imager and does not show a significant improvement in the stability of docking sites. Adding PO+C to the 18-mer spacer (3b) eliminates the depletion of docking sites as effectively, as for the short imager (2a). We hypothesize that this difference is caused by the different irradiances intrinsic to the two applied methods. Photo-reactions from excited states are known to depend non-linearly on the excitation irradiance [19,21,22]. Comparing the feasibility for kinetic investigations compared to localization precision, a large spacer is expected to decrease the accuracy in SMLM, due to the larger accessible space for the fluorophore upon DNA hybridization [9]. On averaged DNA-PAINT images of several hundred origami nanostructures, the 18-mer spacer imager resolves binding sites with overall precision and accuracy (Figure A4) comparable to the imager with the fluorescent dye attached in close proximity to the docking strand. Individually localized origami nanostructures exhibit a significant loss in image quality that is additionally reduced by the lower association rate of the 18-mer spaced imager (Table 1). Interestingly, adding PO+C to the 18-mer spacer (3b) improves the achievable resolution, reaching results similar to the conventional imager strands (2a). We attribute the reduced blur of the 18-mer spacer with PO+C on the averages to the lower association rate and thus fewer simultaneous binding events (Figure A4). The effect is alternatively achieved in DNA-PAINT acquisitions by a lower imager concentration. A reduced association rate of the 18-mer spaced imager compared to the conventional imager can be partially attributed to a larger hydrodynamic radius, as experimentally confirmed by diffusion measurements in confocal FCS (Figure A2c). For applications based on the kinetics of binding, in particular SI-FCS [15] and qPAINT [13], adding a spacer sequence improves the photo-stability at low-irradiance conditions, while being free from chemical modifications and not requiring specialized buffers. Generally, we expect the positioning of the fluorescent dye with respect to the binding sequence to offer unexplored optimization potential for reducing photo-toxicity in applications without the need for maximally precise localization.

In SI-FCS, a second component to the autocorrelation function can be observed at lag times larger than 10 seconds with concentrations of conventional imager strands higher than 30 to 100 nM (Figure A7b). We previously speculated that unspecific binding might be the main cause for this second component [15]. In the light of the results presented in this work, we now assume the photo-induced depletion of binding sites to be the primary cause (Figure A7b). While at low concentrations the applied detrending is sufficient to eliminate additional contributions in the correlation curve, with increasing concentration, the depletion and therefore the second component becomes more pronounced. The oxygen scavenging system PO+C massively reduces the depletion of docking strands even at the highest employed concentrations (300 nM, Figure A7a) and therefore removes the second contribution from the autocorrelation curves (Figure A7b). The second employed oxygen scavenging system (PCD+PCA) exhibits fluctuating intensity traces that are not fully described by a monoexponential detrending. In particular, we observe periods with increasing average fluorescence intensities that are not observed in any other experimental condition. In our hands, PCD+PCA causes less stable experimental conditions, which manifest themselves as fluctuating fluorescence intensities over time. This effect is particularly pronounced for sample life spans exceeding hours or days. The observed instabilities may potentially be attributed to a nuclease contamination of PCD [63]. In case of PO+C, we did not find any hint on alterations of the sample and therefore conclude PO+C to be favorable for long lasting acquisitions. The 18-mer spacer exhibits a second component of the autocorrelation, which is becoming further pronounced by adding PO+C. We speculate that the 18-mer spacer exhibits additional dynamics on the time scale of tens of seconds, which are independent of the concentration and thus different from previously observed additional components in SI-FCS. Under all conditions, we observe at concentrations above 30 to 100 nM, depending on the condition, that the correlation curves do not decay to zero, irrespective of the addition of oxygen scavenging buffers or the extension with the 18-mer spacer. We therefore assume that a non-perfect focus stabilization, fluctuating laser intensities or temperature changes during a 5-h measurement may account for the multiple components on the time scale of min. However, those contributions separate well in time from the decay of interest of the autocorrelation function and therefore do not hinder the quantitative analysis of autocorrelation curves.

In summary, we have shown that in DNA-PAINT super-resolution microscopy, binding sites are damaged by ROS, which are generated from excited fluorescent dyes on hybridized imager strands. Not only does this effect limit the achievable acquisition time of DNA-PAINT, but it also hinders quantitative analysis based on the kinetics of the hybridization reaction. Similarly, long acquisition series in SI-FCS show artifacts generated by the loss of binding sites even though performed at low irradiances to avoid photo-bleaching. DNA-PAINT super-resolution microscopy and SI-FCS measurements benefit from the use of oxygen scavenging buffers, of which PO+C showed best long-term stability. Further, we presented an extended imager strand with an 18-mer spacer that drastically reduces the depletion of binding sites at low irradiances without additives to the sample. In particular for applications exploiting the kinetics of DNA hybridization, not only the addition of oxygen scavenging systems, but also a design placing the dye at larger distance from the docking site, will improve quantitative analysis. We believe that these results will be of general interest for the future design of fluorescence-based, minimally invasive applications of DNA nanotechnology. To our knowledge, this is the first time that the PO+C oxygen scavenging system has been applied to enhance the performance of DNA-PAINT super-resolution microscopy and we believe this system to be of use also in the context of three-dimensional cellular imaging of fixed specimen, which requires extended acquisition times.

## 4. Materials and Methods

### 4.1. Origami Purification

DNA origami nanostructures were synthesized as described previously [11] and subsequently PEG purified. Folded DNA origami structures were mixed 1:1 (*v*/*v*) with 2× PEG purification buffer (PEG-8000 15% (*w*/*v*), 500 mM NaCl, 1× TE buffer), centrifuged for 30 min at 17,900 rcf and 4 °C. The supernatant was removed and the DNA origami resuspended in folding buffer (12.5 mM MgCl_2_, 10 mM Tris, 1 mM EDTA at pH 8.0) by shaking and heating for 5 min at 600 rpm and 30 °C. Previously described steps were repeated two times to increase the purification. Finally, DNA origami nanostructures were stored at −20 °C until use. The assembly of DNA origami nanostructures was confirmed using DNA-PAINT microscopy (Figure 2 and Figure A3). Origami structures exposed the docking sequence 5′-TTATACATCTA-3′, consisting of a TT-spacer followed by nine nucleotides complementary to the imager sequence.

### 4.2. Buffers

For simplicity, we name the used buffers A+ and B+. Buffer A+ contains 10 mM Tris-HCl, 100 mM NaCl, 0.05% (*v*/*v*) Tween20. Buffer B+ contains 5 mM Tris-HCl, 10 mM MgCl_2_, 1 mM EDTA, 0.05% (*v*/*v*) Tween20. Enyme buffer for the PO+C oxygen scavenging systems consists of 10 mM Tris pH 7.5, 50 mM KCl and 20% glycerol for better stability of the stock solutions.

### 4.3. Fluorescent Imager Solutions

Labeled imager strand solutions were used in varying target concentrations (10, 30, 100, 300 or 600 nM) in five conditions:‘Conventional’, as employed previously [11,15] with the sequence 5′-CTAGATGTAT-3′-Cy3B (Eurofins SAM, Ebersberg, Germany) [64];‘Oxygen scavenger’(a)PO+C, as (1) but incubated for 1 hour prior to measurement with the PO+C oxygen scavenger system (1× PO, 1× C, 0.8 % Glucose as described in [65]), with 1× Trolox added. Stock solutions: 100× PO solution consists of 26 mg of PO (P4234-250UN, Sigma-Aldrich Chemie GmbH, Taufkirchen, Germany), 684 µL of enzyme buffer; 100× C solution consists of 2 mg Catalase in 1 ml enzyme buffer. Both were centrifuge filtered (Ultrafree MC-GV, Merck KGaA, Darmstadt, Germany; 0.22 µm), flash frozen in liquid nitrogen and stored at −80 °C; 100× Trolox solution consists of 100 mg of Trolox (Sigma-Aldrich 238813-1G), 430 µL of methanol and 345 µL of NaOH (1 M) in 3.2 mL of H_2_O, stored at −20 °C);(b)PCD+PCA, as (1) but incubated for 1 hour prior to measurement with the PCD+PCA oxygen scavenger system (1× PCD, 1× PCA, 1× Trolox), as described in [11]. Stock solutions: 40× PCA solution consists of 154 mg of PCA (37580-25G-F, Sigma-Aldrich) in 10 mL of water, adjusted to pH 9.0 with NaOH; 100× PCD solution consists of 9.3 mg of PCD (P8279-25UN, Sigma-Aldrich) and 13.3 mL of buffer (50% glycerol stock in 50 mM KCl, 1 mM EDTA and 100 mM Tris–HCl, pH 8.0), Trolox as above, all stored at −20 °C;‘18-mer spacer’,
(a)with the sequence Cy3B-5′-GTT ATG GGT GGT TTG GGG-CTAGATGTAT-3′ (Eurofins SAM), where the hybridizing nucleotide sequence is identical to (1). Incubation 1:1 at identical concentration with 5′-CCC CAA ACC ACC CAT AAC-3′ complementary unlabeled strands forms a stable duplex, increasing the persistence length of the 18-mer spacer. The fluorescent dye is attached at 5′ to maximize the distance of dye and hybridizing nucleotides. The sequence was checked to not form a secondary structure using the Nucleic Acid Package (NUPACK) [66];(b)as in (3a) but with PO+C added as in (2a).

### 4.4. Sample Preparation

Sealed sample chambers were prepared as described previously [11,15]. In brief, high precision #1.5 coverslips (Paul Marienfeld GmbH, Lauda Königshofen, Germany) were sonicated in acetone (chemical grade, Merck KGaA, Germany) for 10 min and then rinsed twice with ethanol (chemical grade, Merck Millipore, Germany) and water (milli-Q, Merck KGaA, Darmstadt, Germany) and gently dried with pressurized nitrogen. The cleaning of the coverslip was completed by putting a drop of 2-propanol on it (Uvasol, Merck KGaA, Darmstadt, Germany) and wiping with a paper tissue (Kimtech Science, Sigma Aldrich). The same procedure was performed on microscope slides (76 × 26 mm², Menzel, Thermo Fisher Scientific, Waltham, MA, USA). The high precision coverslip and the microscope slide were assembled into a flow chamber by gluing them together with double-sided sticky-tape (Scotch, Conrad Electronic SE, Hirschau, Germany), yielding a roughly 5 × 22 × 0.08 mm^3^ large chamber. For DNA-PAINT unspecific binding between origami structures is easily detected in the final image and thus glass surfaces were used without prior cleaning. In a series of volume exchanges, the flow chamber was first incubated with 20 µL of 1 mg/mL albumin, biotin-labeled bovine (Sigma-Aldrich) in buffer A+ for two min, washed with 40 µL buffer A+, incubated with 20 µL of 0.5 mg/mL streptavidin (Thermo Fisher Scientific) in buffer A+ for two min, washed with 40 µL buffer A+ and washed with 40 µL buffer B+.

For DNA-PAINT imaging, 20 µL folded DNA origami solution, diluted 1:200 from PEG purified solution in B+ buffer, were incubated for five min. For SI-FCS, 20 µL folded DNA origami solution, diluted 1:20 from PEG purified solution in B+ buffer, were incubated for ten min. Chambers were washed with 40 µL buffer B+ and finally loaded with 20 µL of imager solution in the required condition (10 µM for DNA-PAINT imaging). In a final step, the chamber was sealed using two-component epoxy glue (Toolcraft, Conrad Electronic SE) or picodent twinsil 22 two component glue (picodent, Wipperfuerth, Germany). We verified the final concentration of fluorescently labeled ssDNA by confocal FCS measurements.

### 4.5. DNA-PAINT Microscopy Setup

DNA-PAINT imaging was carried out on an inverted custom-built microscope in an objective-type TIRF configuration with an oil-immersion objective (UAPON, 100×, NA 1.49, Olympus Europe, Hamburg, Germany). Fluorophores were excited with a DPSS laser with a wavelength of 561 nm (output power 1W, DPSS-system, MPB Communications Inc., Montreal, QC, Canada). Laser power was adjusted by polarization rotation with a half-wave plate (WPH05M-561, Thorlabs, Dachau, Germany) before passing a polarizing beam-splitter cube (PBS101, Thorlabs). To spatially clean the beam-profile the laser light was coupled into a single-mode polarization-maintaining fiber (P3-488PM-FC-2, Thorlabs) using an aspheric lens (C610TME-A, Thorlabs). The coupling polarization into the fiber was adjusted using a zero-order half wave plate (WPH05M-561, Thorlabs). The laser light was collimated after the fiber using an achromatic doublet lens (AC254-050-A-ML, Thorlabs) resulting in a collimated beam of ~6 mm full-width at half-maximum (FWHM). The laser beam was magnified by a factor of 2.5 using a telescope custom-built from two achromatic doublets (AC254-030-A-ML and AC508-075-A-ML, both Thorlabs). The excitation light was finally focused in the objective’s back focal plane using an achromatic doublet lens (AC508-180-A-ML, Thorlabs). Fluorescence light was separated from the excitation by a dichroic beam splitter (F68-785, AHF Analysentechnik, Tübingen, Germany) directly below the objective, spectrally filtered with an emission filter (605/64, AHF Analysentechnik). The signal from the sample was finally imaged on a sCMOS camera (Zyla 4.2, Andor Technologies, Belfast, UK) without further magnification (TTL180-A, Thorlabs) resulting in an effective pixel size of 130 nm (after 2 × 2 binning). Microscopy samples were mounted on an x-y-z stage (S31121010FT and FTP2050, both Advanced Scientific Instrumentation, Eugene, OR, USA) used for focusing and laterally moving the sample with the microscope objective fixed in position.

### 4.6. DNA-PAINT Image Acquisition

DNA-PAINT super-resolution data was acquired with a sCMOS camera using µManager [67]. Acquisition parameters were: full chip 2 × 2 pixel binning, read out rate 200 MHz and dynamic range 16-bit. The exposure time was set to 200 ms, resulting in a camera frame rate of 5 Hz and in ~83 min of total measurement time. The excitation power was set to 22 mW behind the objective with a Gaussian shaped illumination with a $1/e^{2}$-width of 84 µm, resulting in a peak irradiance of $0.2\;{\mathrm{kW}/\mathrm{cm}}^{2}$.

### 4.7. DNA-PAINT Data Analysis

Super-resolved DNA-PAINT images were computed with Picasso according to [11]. Binding events were localized by Gaussian least-square fitting with a net gradient chosen to suppress localizations of noise. Localizations were drift corrected in a three-step process, first by redundant cross-correlation (RCC) [11,68], subsequently based on picked DNA origami and finally based on picked individual docking sites. To generate subsets, the drift corrected localizations were split in subsets of 5000 frames each with a custom-written Python script and finally rendered individually. Super-resolved images were rendered blurring individual spots based on the global localization precision. To analyze bright and dark times, origami structures were picked (Picasso: ‘pick similar’) and further analyzed by a custom-written Python script (https://github.com/DerGoldeneReiter/qPAINT) acting on Picasso’s ‘Picked localization’ files. Dark times of one frame were ignored to reduce artifacts caused by single missed localizations. For docking site analysis picked origami structures were averaged (translation and rotation) to a designed model structure using the ‘average3’ module of Picasso with a pixel oversampling of 40, setting a custom symmetry of 180 degrees [46]. All individual docking sites were picked on the average image (Figure A4) in the render module of Picasso. ‘Unfold’ translates the picks of the average back to the individually picked structures and thus, picks of the individual docking sites on every origami structure are obtained. After counting the localizations of each picked single docking site, a lower cut-off value (three localizations) was used to identify active individual docking sites. Dividing the number of the active dockings sites by the number of the originally picked origami structures used for averaging yields the average number of docking sites per origami.

### 4.8. SI-FCS TIRF Microscope

DNA-PAINT and SI-FCS time series were recorded on a custom-built TIRF microscope, constructed around a Nikon Eclipse Ti-S body as described previously [15]. Fluorescence was excited by 561 nm diode-pumped solid state (DPSS) laser (Cobolt Jive, 50 mW nominal, Hübner GmbH & Co. KG, Kassel Germany), spatially filtered by a single-mode fiber (kineFLEX-P-3-S-405.640-0.7-FCS-P0 and kineMATIX, Qioptiq, Hamble, UK), collimated (f = 25 mm, all standard achromats, Edmund Optics, Karlsruhe, Germany), linearly polarized (CCM1-PBS251/M, Thorlabs) and three-fold magnified (f = −25, 75 mm). The TIRF angle was controlled by translating the focus (f = 225 mm) of the excitation beam by means of a piezo-electric stage (Q545, Physikalische Instrumente, Karlsruhe, Germany) in the back focal plane of the objective (Nikon SR Apo TIRF, 100× magnification, 1.49 numerical aperture (NA), Nikon, Düsseldorf, Germany).

Fluorescence emission was separated from the excitation (zt405/488/561/640rpc flat, AHF Analysentechnik, Tübingen, Germany) before entering the microscope body. The image on the camera-port was relayed on an electron-multiplying charge-coupled device (EMCCD) camera (iXon Ultra 897, Andor Technologies) by an additional 4f telescope (f = 200 mm, AC254-200-A-ML, Thorlabs). Laser emission was attenuated and synchronized with the camera acquisition by an acousto-optical tunable filter (TF-525-250, Gooch & Housego, Torquay, UK), which was interfaced through a PCI Express card (PCIe-6323 and BNC-2110) and controlled with a custom LabView 2011 software (all National Instruments, Austin, TX, USA). The emission light was additionally band-pass filtered (593/46).

Drifting of the focus position was eliminated by a custom-built focus stabilization. A near-infrared laser (LP785-SF20, Thorlabs) was totally internally reflected from the glass-water interface of cover-slide and sample. The beam position was monitored on a CMOS camera (UI-3240CP-NIR-GL, Imaging Development Systems, Obersulm, Germany). A feedback control implemented in LabVIEW 2015 (National Instruments) maximized the cross-correlation of the images of the laser spot and a reference image, respectively. The axial sample position was adjusted every 200 ms accordingly (P737.2SL and E-709.SRG, Physikalische Instrumente). The sample and objective were temperature stabilized to 23 °C. (H101-CRYO-BL stabilization unit, with H101-MINI sample chamber and OKO-MOC objective stabilization, Okolab, Ottaviano, Italy).

### 4.9. SI-FCS Image Acquisition

Images were recorded using the Andor Solis software (Andor Technologies, Version 4.28) with 4 × 4 hardware binning as 64 × 64 pixel images for 1.5 million frames, as described previously [15], resulting in ~5 h total measurement time. The exposure time was 10 ms, resulting in a camera frame rate of 85 Hz. The excitation power was set to 0.75 mW behind the objective, with a Gaussian shaped illumination with $a\;1/e^{2}$-width of 51 µm, resulting in a peak irradiance of 0.018 kJ/cm^2^. The EMCCD camera was used with electron multiplying gain, adapted according to the brightness of the sample.

### 4.10. SI-FCS Data Analysis

The autocorrelation curves were computed and analyzed using a custom-written Matlab 2017a (The MathWorks, Natick, MA, USA) software, described previously [15]. Intensity traces were generated by additional 8 × 8 software binning, resulting in 5.12 µm effective integrated area size, representing 32 × 32 native camera pixel. The signal in each pixel was integrated, yielding 64 intensity traces, which were bleach and drift-corrected by a single exponential, and individually correlated using the multiple-τ algorithm [69], in which we doubled the bin width after every sixteenth point in the autocorrelation curve. The obtained autocorrelation curves were fitted individually by a single exponential decay with an offset, from which the amplitude and the characteristic decay time were obtained. Similar to [15], concentrations above 100 nM were fitted with a bi-exponential to account for the second component originating from depletion of docking sites.

### 4.11. Direct Measurement of the Concentration of Imager Strands with Confocal FCS

We measured the solution concentrations of imager strands with confocal FCS, as described previously [15]. In brief, we used a commercial LSM 780 ConfoCor3 system (Zeiss AG, Oberkochen, Germany) with the confocal volume positioned 40 µm above the cover slide. We calibrated the confocal volume using Alexa546NHS (Thermo Fisher) and its reported diffusion coefficient $D=341\;\frac{{µm}^{2}}{s}$ at 22.5 °C [70]. We calculated the corresponding diffusion coefficient at the measurement temperature (26.5 °C to 27 °C) using the well-known relation $D\sim \frac{T}{\eta \left(T\right)}$ and an empirical expression for the temperature dependence of the viscosity η of water [71]. We applied a simple 3D diffusion model function:(7)$$G\left(\tau \right)=N^{-1}{\left(1+\frac{\tau }{\tau_{D}}\right)}^{-1}{\left(1+\frac{\tau }{S^{2}\tau_{D}}\right)}^{-\frac{1}{2}},$$
as justified previously [15]. Here, N is the average number of particles in the detection volume, $\tau_{D}=\frac{w_{xy}^{2}}{4D}$ the diffusion time, with $w_{xy}$ being the $e^{-2}$-value of the Gaussian detection volume and S the structure parameter. Concentrations are directly obtained from the amplitude of the correlation curves: $c=N{\left(\pi^{\frac{3}{2}}w_{xy}^{3}S\right)}^{-1}$. As diffusion coefficient, we measured $D_{\mathrm{conv}}=\left(201\pm 5\right)\frac{{µm}^{2}}{s}$ (n = 8), in agreement with previously reported results [15,72] and $D_{18-\mathrm{mer}\;\mathrm{spacer}}=\left(120\pm 20\right)\frac{{µm}^{2}}{s}$ (n = 4) for the 18-mer spacer. The presented numbers correspond to mean and standard deviation of the indicated n measurements, each of them at least 20 min long.

### 4.12. Simulation of Bright and Dark Times

Simulations of the bright and dark time distributions were generated with COPASI [73]. Biochemical parameters for the simulation were: Mass action irreversible (A+B→C, C→A+B), $k_{a}=1.5\cdot 10^{6}\;{\left(\mathrm{Ms}\right)}^{-1}$, $k_{d}=0.3\;s^{-1}$ with varying number of initial species values (B = 12,10,8,6,4) corresponding to single dockings sites and fixed imager concentration of 〈A〉= 10 nM. Time course simulations were performed with the same interval spacing (0.2 s) and total acquisition time (17 min per time segment) used in the DNA-PAINT image acquisitions. Obtained traces were analyzed analogous to experimental data using a custom-written Python script (https://github.com/DerGoldeneReiter/qPAINT).

## Figures and Tables

**Figure 1 molecules-23-03165-f001:**
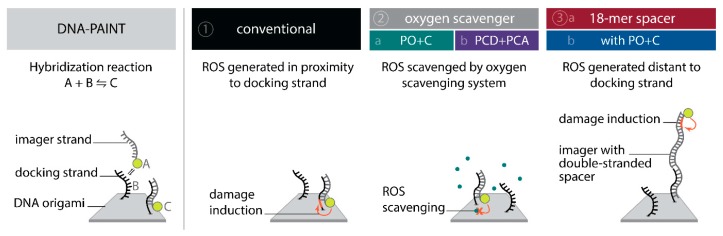
Schematic of DNA-PAINT with the binding of imager strands (A) to docking strands (B) forming the hybridized duplex (C) (left panel). Overview of the conditions to explore the photo-induced depletion of binding sites in DNA-PAINT microscopy and kinetic measurement with SI-FCS (right panel): (**1**) conventional 9 nt overlapping imager, (**2**) identical imager as in (1) but in presence of a oxygen scavenging system (2a: PO+C or 2b: PCD+PCA), (**3a**) a modified imager with identical binding sequence and buffer conditions as in (1) but extended by an double-stranded spacer, increasing the distance of dye and docking strand and (**3b**) the combination of 18-mer spacer and PO+C. ROS scavenging and an increased distance of the fluorescent dye to the docking strand are ought to decrease the rate of depletion of docking sites.

**Figure 2 molecules-23-03165-f002:**
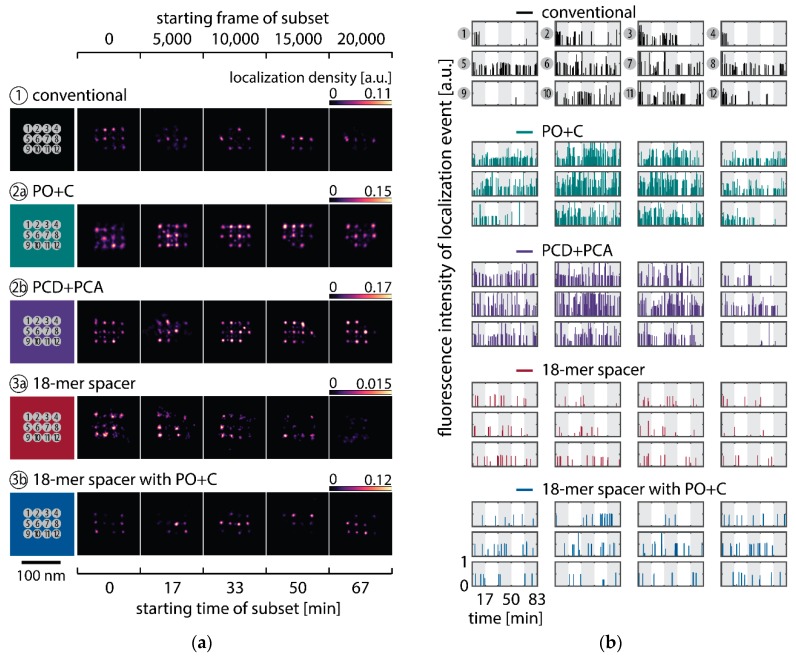
Long DNA-PAINT acquisition of DNA origami nanostructures with 12 exposed docking strands arranged in a 3 × 4 grid with 20 nm spacing. Five conditions are displayed: conventional imager (1, black), conventional imager with oxygen scavenging system added (2a: PO+C, turquoise; 2b: PCD+PCA, purple), imager with 18-mer spacer between docking sequence and fluorescent dye (3a, red) and the 18-mer spacer with PO+C (3b, blue). (**a**) Time series of representative super-resolved DNA-PAINT images, reconstructed from five subsequent 5,000 frame long subsets of a 25,000 frame long acquisition (in total 83 min). Additional examples are listed in the Appendix B (Figure A3). Scale Bar: 100 nm. (**b**) Time traces of localizations within circular areas picked as individual bindings sites. The alternating shade of the background indicates the five subsets. DNA-PAINT raw data is available as Appendix A.

**Figure 3 molecules-23-03165-f003:**
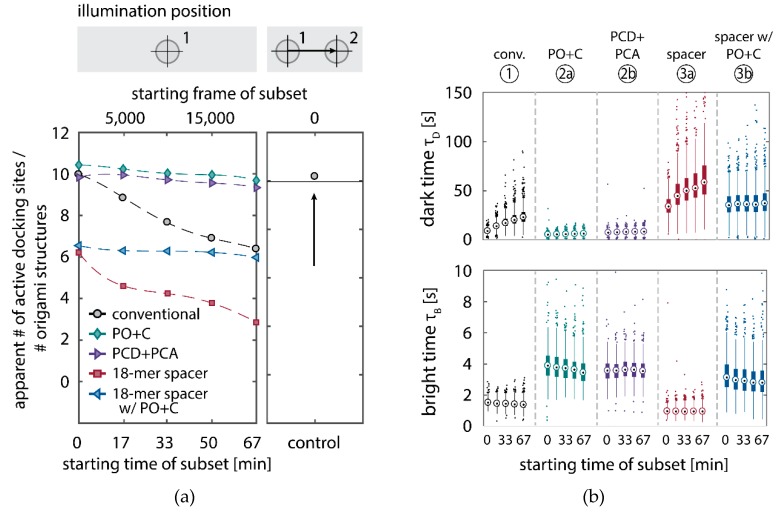
Quantitative analysis of the depletion of docking strands in DNA-PAINT super-resolution microscopy. The five conditions displayed are identical to Figure 2. (**a**) Left panel: active docking sites are counted individually on DNA origami nanostructures, based on the back-translation of the position of docking sites picked on automatically averaged nanostructures (Figure A4) and divided by the total number of identified origami structures. The dashed line is a guide to the eye. Right panel: a control of 5000 frames, equivalent to the first subset, was measured in a previously not irradiated area on the same sample and compared to the initial number of active docking sites (solid horizontal line) for the conventional condition (right panel). The total numbers of identified DNA origami nanostructures for the five conditions are: (1) 786, (2a) 824, (2b) 566, (3a) 690, (3b) 580 and 690 for the control of condition (1). (**b**) Box plots of the bright times $\tau_{B}$ and dark times $\tau_{D}$ for the subsets shown in (a). Circles indicate the median; bottom and top edges of the box (bold vertical lines) indicate the 25th and 75th percentiles, respectively. The whiskers extend to the data points not considered outliers (thin vertical lines); outliers are plotted individually as dots.

**Figure 4 molecules-23-03165-f004:**
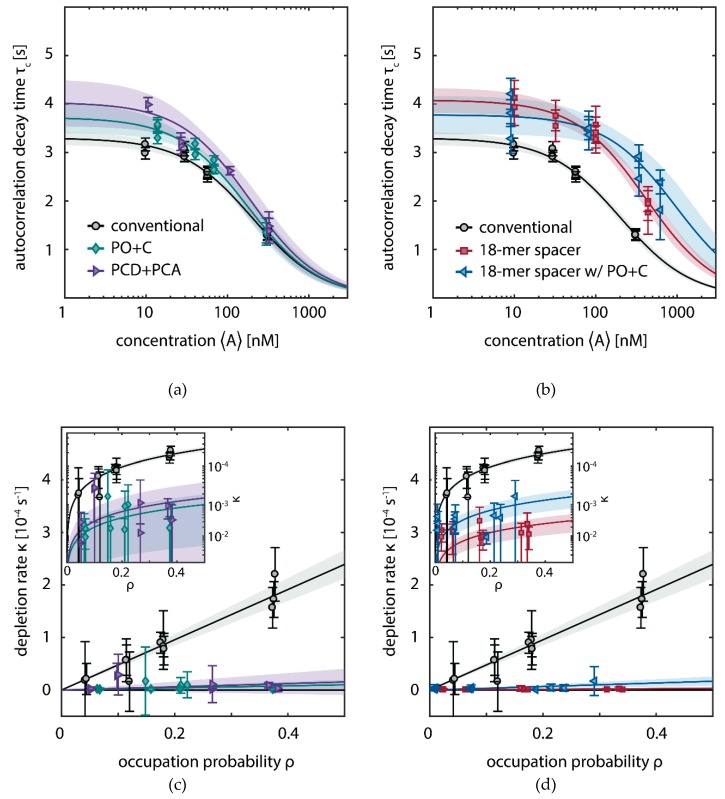
Depletion of docking sites investigated by SI-FCS at low irradiance ($0.018\;\mathrm{kW}/{\mathrm{cm}}^{2}$). (**a**) Titration series of the imager concentration 〈A〉 with 9 nt overlap for conventional imager strands (black), addition of oxygen scavenging system PO+C (turquoise) and PCD+PCA (purple). Data points and error bars represent mean and standard deviation from 64 regions of interest (5.1 × 5.1 µm), respectively. Solid lines show the fit to Equation (2). The 95% functional error bounds are displayed as shaded areas. (**b**) As (a) but comparing the conventional imager to the 18-mer spacer (red) and 18-mer spacer with PO+C (blue). (**c**) The effective depletion rate κ (Equation (1)) of docking strands depends linearly on the occupation probability ρ, which is calculated from the concentration and the kinetic rate constants according to Equation (6). Solid lines show the fit to Equation (4). Data points and errors are displayed as described in (a). The inset shows the depletion rate κ on a logarithmic scale. (**d**) Samples as in (b) displayed as in (c). SI-FCS raw data is available as Appendix A.

**Table 1 molecules-23-03165-t001:** Binding kinetics as measured by SI-FCS titrations of the imager concentrations 〈A〉. Hybridization rates and affinity as obtained from the titrations in Figure 4. The errors represent the 95% confidence bounds of the fit.

Imaging Condition	$k_{a}\;\left[10^{6}\cdot {\mathrm{Ms}}^{-1}\right]$	$k_{d}\;[s^{-1}]$	$K_{d}\;[\mathrm{nM}]$
conventional	1.49±0.17	0.303±0.010	200±30
PO+C	1.5±0.4	0.268±0.017	180±60
PCD+PCA	1.3±0.4	0.25±0.02	190±70
18-mer spacer	0.59±0.11	0.245±0.012	420±100
18-mer spacer with PO+C	0.28±0.12	0.26±0.02	1000±500

**Table 2 molecules-23-03165-t002:** Slope of the linear fit of Equation (4) to the depletion rate κ(ρ) in Figure 2. The errors represent the 95% confidence bounds of the fit.

Imaging Condition	$\delta \;\left[10^{-6}\cdot s^{-1}\right]$
conventional	480±55
PO+C	20±19
PCD+PCA	30±50
18-mer spacer	7±4
18-mer spacer with PO+C	33±18

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
