# Peer review of "Photo-Induced Depletion of Binding Sites in DNA-PAINT Microscopy"

_molecules, 2018, doi:10.3390/molecules23123165_

Round 1
Reviewer 1 Report
This is a nice study, which will surely prove useful to scientists doing DNA-PAINT microscopy. The combination of quantification of single-site binding, bright and dark times, and SI-FCS makes it comprehensive and convincing.
Minor points:
I don’t agree that “The limited photon budget of fluorescent dyes is the main limitation for resolution …” That is only the limitation for localization precision. Rather, in dSTORM resolition is limited by a combination of irreversible transitioning to the dark state (photobleaching without being detected) and the non-exponential (long-tailed) distribution of blinkings, that leads to under-sampling of some sites while others are over-sampled. In PALM, it could be the limited photon budget, but also the failure of FPs to fold into a functional label. Once enough photons are collected (O(1000)), the size of the label also becomes important for nearly all targeting methods.
Issue with reference 7 (first and last names reversed)
Strange sentence:
The first effect is usually negligible in DNA-
117 PAINT, as the reservoir of fluorophores in solution is large enough not to affect the concentration of
118 fluorescent dye by imaging a small volume via total internal reflection fluorescence (TIRF)
119 illumination.
The first part makes sense, but the part after “by imaging …” doesn’t fit with it.
Also strange:
Taking into account the fast diffusion of imager strands (determined as ?=_(_2_0_1_±
120 5_)_μm_2_s_⁄, Figure A2), their concentration can be assumed equilibrated and constant throughout the
121 sample.
In any equilibrated solution, one assumes the concentration is approximately constant. But giving the diffusion coefficient as a justification is odd. One might compare D to the length and time scales over which the diffusion of imager strands is relevant, for example the distance between strands at a given concentration and the time scale of a single image.
Figure 2b, the corner spots (1,4,9,12) appear more affected/less labeled than the central ones. Is this just a coincidence in these particular examples? According to the ROS hypothesis presented, there could be a dependence of site position on the concentration of ROS and therefore inactivation/damage.
Line 192-193 Coincident binding of multiple strands is mentioned. It would be nice to go back to events with long bright times, to see whether one could detect single-step unbinding events (the PAINT equivalent of single-step photobleaching).
There are a few typos (concentration is misspelled in Fig. 4a). PAINT is “points accumulation” not “point accumulation.”
Author Response
Please find our answers to the reveiwer's comments in the attached Word document

Reviewer 2 Report
Dear editor,
The manuscript written by Blumhardtet al describe the use of ROS scavengers to avoid photo-induced depletion of binding sites in DNA-PAINT microscopy. Presentation of the data is sound and the paper is written well. Furthermore, the observations may be useful for scientists in the field of super resolution imaging. I recommend accepting this paper and take notions that I list below.
Remarks
Sentence 94: four different conditions are mentioned, but these are also mentioned in the figure legend of Figure 1. This figure is so self-explaining including the figure legend that I would remove the text sentence 94-104.
I would stress that the 18-mer spacer was used in same buffer as conventional conditions. Did the authors test the 18-mer in combination with ROS scavengers? If so, please include in this ms.
If I check Figure 2A, it looks like that the 18-mer has more spots compared to the conventional conditions. However, sentence 166 they mention that lower number of localizations are observed. Furthermore, the structures of the 18-mer look larger compared to other conditions, can the authors comment on these observations?
At the end of the ms, the authors propose the use of ROS scavengers for in vivo applications. I would like to see suggestions how to implement this, addition of Trolox or co-expression of GO? This may be incorporated in the discussion section.
Author Response

(The authors gave the same response as above.)

Reviewer 3 Report
In this manuscript, these authors described photo-induced damage for DNA in a super-resolution microscope. They checked the prevention of the damage by using photo scavengers or longer DNA arms; then, such improvement can reduce the rate of photodamage on DNA. I agree it can improve the quality of super-resolution microscopy and artificial site-specific depletion of DNA origami. The manuscript is written well, especially figure composition, main text, and supplemental results. But, this reviewer has a little bit concerns as follows.
1. In Figure 3b, the bright time tauB in addition of PO+C is slightly decreased as the authors mentioned. What is a possible mechanism on the decrease? Moreover, the decreasing rate seems to be very slight. Is it emphasized to be decreased isn’t it?
2. In Figure 4b, the fitted lines for three kinds (blue, purple, and brown lines) including dashed lines are hard to be distinguished. Their appearance should be improved.
3. Why did these authors try to check the phenomena of the 18-mer spacer using scavengers (PO + C)? I suppose it is a worth to be checked.
Author Response

(The authors gave the same response as above.)

Round 2
Reviewer 3 Report
I completely satisfied. I appreciate the author's revision.